# Improved Extraction Efficiency and Antioxidant Activity of Defatted Canola Meal Extract Phenolic Compounds Obtained from Air-Fried Seeds

**DOI:** 10.3390/antiox11122411

**Published:** 2022-12-06

**Authors:** Olamide S. Fadairo, Ruchira Nandasiri, Thu Nguyen, N. A Michael Eskin, Rotimi E. Aluko, Martin G. Scanlon

**Affiliations:** 1Food and Human Nutritional Sciences Department, University of Manitoba, Winnipeg, MB R3T 2N2, Canada; 2Richardson Centre for Food Technology and Research, 196, Innovation Drive, Winnipeg, MB R3T 2N2, Canada; 3St. Boniface Hospital Albrechtsen Research Centre, 351, Tache Avenue, Winnipeg, MB R2H 2A6, Canada

**Keywords:** canola meal, air frying, phenolic compounds, antioxidant potential, pre-treatment

## Abstract

This study investigated the efficacy of roasting pre-treatment by air frying to enhance the extraction and recovery of the predominant sinapic acid derivatives (SADs) from roasted canola meal and the antioxidant potential of the methanolic extracts. Canola meal was obtained by air frying canola seed at 160, 170, 180 or 190 °C for 5, 10, 15 or 20 min. Oil was extracted using the Soxhlet method, and the de-oiled meal fraction was air-dried. Phenolic compounds were isolated using ultrasound-assisted extraction with 70% (*v*/*v*) methanol and then quantified by high-performance liquid chromatography-diode array detection. The antioxidant potential of the defatted meal methanolic extracts was evaluated using 2,2-diphenyl-1-picrylhydrazyl (DPPH), ferric reducing antioxidant power (FRAP), and metal ion-chelating activity (MIC) assays. The highest total phenolic content of 3.15 mg gallic acid equivalent/g dry weight was recorded in the defatted meal extract from seeds pre-treated with air frying at 190 °C for 15 min. Sinapine, sinapic acid and an unknown compound at a retention time (RT) of 26.6 min were the major sinapates identified in the defatted meal with the highest concentrations of 7572 ± 479.2 µg/g DW, 727 ± 43.45 µg/g DW and 1763 ± 73.5 µg/g DW, respectively, obtained at 160 °C for 5 min. Canolol (151.35 ± 7.65 µg/g DW) was detected after air frying at a temperature of 170 °C for 20 min. The FRAP and MIC correlated positively (r = 0.85) and generally decreased with increased air frying temperature-time conditions. The highest FRAP and MIC values of 0.53 mM and 80% were obtained at 160 °C for 5 and 20 min, respectively. The outcome of this study will contribute new knowledge that could improve the value addition and by-product utilization of canola seeds.

## 1. Introduction

Canola (*Brassica napus* L.) is among the leading oilseed crops globally [1]. A large quantity of co-products, such as defatted meal or cake, are often generated during the oil extraction process owing to the high demand for canola oil. These co-products generally find limited applications as feed ingredients in the livestock industry, and hence are under-exploited [2,3]. Canola meal is an excellent source of valuable functional bioactive compounds, including amino acids and phenolics [2]. The defatted meal comprises free and bound phenolics. Sinapine, a precursor of canolol, constitutes more than 80% of the total phenolics [4,5]. In addition to sinapine, canola meal contains sinapic acid and some flavonoid derivatives [5,6,7,8]. Processing conditions such as elevated temperature and pressure impact the recovery and amount of phenolic compounds [9,10]. Of particular interest are the relatively new polyphenolic compounds identified as canolol, its dimers, oligomers, and other degradation derivatives with enhanced antioxidative properties that areoften generated under elevated temperature and pressure [10]. Moreover, the extractability of some predominant sinapic acid derivatives can be improved by subjecting the seeds to increased temperature (∼200 °C) and pressure (∼2000 psi) processing conditions [10].

Canolol, an important vinyl phenol derivative, has potent antioxidant, anticancer, and lipid-lowering properties, with a potential for the prevention and treatment of cardiovascular diseases [11,12,13] with potential applications in both the food and healthcare industries. While there is a great abundance of polyphenols in canola/rapeseed, a large proportion abounds in the residual cake, and only a relatively low amount is retained in the oil following seed pressing and oil extraction [6]. Canolol formation was first recorded in the oil fraction of rapeseed at elevated temperatures and in deodistillates during oil refining [14]. Subsequent research showed that this sinapic acid derivative could also be generated from the co-products of oil refining, such as the defatted meal [15]. Hence, new processing techniques for generating canolol formation from oil-seed co-products are being investigated as a value-added co-processing strategy. A few studies have documented canolol production from the meal fraction of rapeseed/canola using a variety of pre-treatment and extraction techniques. A recent study by Nandasiri et al. [7] used green pre-treatments, including a modified RapidOxy^®^ 100 equipment combined with ultrasound-assisted extraction, to effectively produce canolol from canola meal. The highest canolol contents of 453 ± 17.66 μg/g dry weight (DW) and 427.1 ± 7.12 μg/g DW at pre-treatment time-temperature regimens of 160 °C and 180 °C for 5 and 10 min, respectively, have been reported [7]. Zago et al. [16] used microwave roasting as another green method and reported that high-temperature steam (105–160 °C) and microwave temperature (160–180 °C) were effective methods for canolol recovery from defatted canola meal. Zago et al. [16] concluded that microwave roasting at 160 °C generated around a 6-fold increase in canolol compared to superheated steam at 180 °C. An earlier study by Niu et al. [17] reported that roasting canola meal via microwave exposure for 7 min reduced the sinapine content by 17% while yielding a 6-fold increase in the canolol content.

While there have been previous studies using a variety of roasting techniques to improve the production of canolol and other major meal-derived phenolics, this is the first study on the application of air roasting techniques for generating canolol and other sinapates from canola meal. Therefore, the objective of this study was to valorize defatted canola meal using air frying of the seeds as a green pre-treatment method followed by an ultrasound-assisted extraction technique to produce canolol and other polyphenols as natural antioxidants. This combined approach will help to establish the optimum extraction efficiency of canolol and other sinapates from defatted canola meal while minimizing the environmental impact. This will also enhance the value-added processing and by-product utilization of defatted canola meals as an additional source of revenue for the canola industry.

## 2. Materials and Methods

### 2.1. Roasting of Canola Seeds Using Air Fryer

Canola seed roasting was carried out by the modified method of Fadairo et al. [18]. Canola seeds were weighed (15.0 g) into an aluminum dish, spread uniformly, placed in a pre-heated air fryer (T-Fal XL, Model number EY201D50, Scarborough, ON, Canada) and roasted at 160, 170, 180, or 190 °C for 5, 10, 15 or 20 min. Cooling of the roasted seeds was performed in a desiccator at room temperature (25 °C), after which they were ground into fine flour with a coffee grinder and used for subsequent analysis.

### 2.2. Preparation of Canola Meal

Defatting of the air-fried canola seed with hexane was performed following the method of Khattab et al. [3] with slight modifications as described by Fadairo et al. [18]

### 2.3. Ultrasonic Extraction of Defatted Canola Meal

The extraction of phenolics from defatted canola meal was done according to the method of Liang et al. [19].

### 2.4. HPLC Analysis of Canola Meal-Derived Sinapic Acid Derivatives

The changes in the phenolic composition of defatted canola meal methanolic fractions extracted by ultrasound-assisted extraction were evaluated by high-performance liquid chromatography (HPLC) as described by Nandasiri et al. [10]. The major phenolic compound, canolol, was identified using the authentic standard with purity of 98% and a detection limit of 0.001 mg/mL. The calibration curve for each standard compound was obtained from 1.0 to 100 μg/mL (*n* = 11) concentration range with R^2^ = 0.9994 for sinapic acid, R^2^ = 0.9999 for canolol, and R^2^ = 0.9996 for sinapine. Sinapic acid (purity > 98%), gallic acid ≥ 98%, HPLC grade solvents and other analytical reagents were purchased from Fisher Scientific Canada Ltd (Ottawa, ON, Canada). Sinapine (purity > 97%) and canolol (purity > 97%) were purchased from ChemFaces® Biochemical Co., Ltd. (Wuhan, Hubei, China). A flow chart for the extraction of defatted canola meal-derived sinapates is shown in Figure 1.

### 2.5. Determination of Total Phenolic Content (TPC)

The TPC of defatted canola meal extracts was determined according to the Folin–Ciocàlteu method described by Cai et al. [20] with a few modifications. A 500 µg/mL gallic acid stock solution was prepared with 70% (*v*/*v*) methanol. A calibration curve was prepared from a series of standard solutions (25–350 µg/mL) using the gallic acid stock solution with linearity (R^2^ = 0.9987).

### 2.6. Evaluation of Antioxidant Properties

#### 2.6.1. 2,2-Diphenyl-1-picrylhydrazyl Radical Scavenging Activity

The radical scavenging activity of the meal extracts was determined according to the method of Girgih et al. [21] with slight modifications for a 96-well flat bottom plate.

#### 2.6.2. Ferric Ion Reducing Antioxidant Power

The ferric ion reducing antioxidant power (FRAP) was carried out according to the method described by Benzie and Strain [22] with modifications for a 96-well plate reader.

#### 2.6.3. Metal-Ion Chelation Properties

The metal ion chelating ability was evaluated according to the modified method described by Xie et al. [23].

### 2.7. Statistical Analysis

Results were expressed as means ± SD (*n* = 3). The differences between mean values were determined by analysis of variance (ANOVA). The treatment conditions were temperature (160, 170, 180 and 190 °C) and time (5, 10, 15 and 20 min), respectively. All data analysis and visualization were done in R 4.1.3 (R Core Team, 2013) and R studio 2021.09.0. All tests were considered statistically significant at *p* < 0.05. Correlation analysis of the predominant sinapic acid derivatives and related antioxidant activity was further performed using R 4.1.3 (R Core Team, 2013) and R studio 2021.09.0 to determine the relationship between the phenolic derivatives and the various antioxidant activities evaluated.

## 3. Results and Discussion

### 3.1. Phenolic Composition of De-Oiled Canola Meal as Affected by Air Frying

The effect of seed roasting by air frying on the phenolic composition of defatted canola meal is shown in Table 1. Sinapine was the most abundant phenolic compound, but the content decreased with an increase in air frying temperature and time (Table 1).

Air frying conditions of 160 °C for 5 and 10 min, respectively, produced the highest sinapine content (7571.6 ± 479.2 and 7471 ± 197.5 SPE µg/g DW) with no significant difference (*p* > 0.05) between them. Moreover, sinapine was also recovered (7193 ± 86 µg/g DW) after 20 min of roasting at 160 °C. Conversely, the lowest amount of defatted meal sinapine (5940.32 ± 115.14 µg/g DW) was obtained with a seed pre-treatment temperature of 170 °C for 20 min. Sinapine, sinapic acid and canolol are the major heat-labile sinapic acid derivatives (SADs) in the defatted canola meal, and their extractability and concentration have been reported to be dependent on the pre-treatment and processing conditions employed [10,24,25,26]. It is evident from Table 2 that sinapine correlated positively with sinapic acid and RT-26.6 min compound (r = 0.75 and 0.93), respectively.

A concomitant increase in the concentration of these sinapic acid derivatives was observed with decreasing air frying temperature and time (Table 1). In contrast, sinapine showed a strong negative correlation with canolol (r = −0.79). The observed relationship was evident by the low canolol content, while the higher sinapine and sinapic acid contents obtained in this study are consistent with the two-step process for canolol formation in which sinapine was first converted to sinapic acid, followed by its subsequent decarboxylation to canolol.

The highest sinapine value observed in the current study is similar to the results of Nandasiri et al. [7], who recorded the highest values of 7641.86 ± 35.75 and 7622.22 ± 181.27 μg/g DW at pre-treatment temperatures of 120 and 180, respectively after 2 min using an inert pressured heating system of a modified RapidOxy 100. In the same study, they reported a 53% reduction of sinapine content when the seed pre-treatment time was increased from 2 min to 20 min at 180 °C. Zago et al. [16] studied the influence of different treatments on rapeseed meal (similar to canola meal) phenolics. They concluded that simple seed roasting pre-treatment had no effect on sinapine content of the meal, but the content decreased by 13% when subjected to superheated steam processing. Wang et al. [27] recently documented the highest quantity of sinapine (12031.96 mg/kg, 14168.67 mg/kg, and 9037.28 mg/kg, respectively in *Brassica napus*, *Brassica rapa* and *Brassica juncea* canola varieties) under steam explosion pretreatment. Khattab et al. [25] reported sinapine as the main phenolic in rapeseed meal (1–2 % *w*/*w*) and a significantly lower sinapic acid content while Yang et al. [28] reported sinapine as the predominant phenolic compound (459.22 mg/100 g) in untreated rapeseed. However, there was a reduction in sinapine content after 8 min of microwave roasting. The authors attributed the decreases to the partial breakdown of sinapine as a result of increased temperature produced by long microwave treatment. Blair & Reichert (1984) [29] reported the concentration of sinapic acid in canola flour, industrial meal, and a protein isolate as 0.39, 0.24, and 0.20 mg/g, respectively. Liu et al. [30] found the sinapine content of defatted rapeseed hulls and dehulled flours from 10 rapeseed varieties ranged from 93 to 176 mg/100 g and 1565 to 2188 mg/100 g, respectively. They attributed the variations in the sinapine levels to differences between cultivars, climatic conditions, growth conditions, degree of maturity, and method of analysis adopted. Similar results were found by Nandasiri et al. [31] indicating other than the sinpates, kaempferol derivatives also contributes to the total phenolic content and its antioxidant activity.

The changes in the sinapic acid concentration in canola meal fractions obtained by air frying pre-treatment of the seeds is shown in Table 1. Interestingly, sinapic acid content followed a similar trend as sinapine under the various air frying conditions. Generally, the concentration of sinapic acid decreased with increase in temperature-time combinations. The highest sinapic acid content (726.88 ± 43.45 µ/g DW) was obtained at roasting condition of 160 °C for 5 min while the lowest sinapic acid content (214.21 ± 16.8 µ/g DW) was recorded at 190 °C for 20 min. The maximum level of sinapic acid observed in this study was higher than that reported by Nandasiri et al. [7]. Wang et al. [27] reported the maximum amount of sinapic acid in *Brassica napus* rapeseed following steam explosion pre-treatment was 1210 mg/kg. In a study by Yang et al. [28], sinapic acid was reported as the main phenolic acid in rapeseed with a value of 89.89 mg/100 g after 7 min of microwave pre-treatment. These results are similar to the findings by Krygier et al. [32] who reported sinapic acid contents of 73.2− 80.1 mg/100 g in rapeseed hull. Kozlowska et al. [33] also obtained sinapic acid concentrations of 41.3−51.6 mg/100 g in rapeseed while a sinapic acid content of 2.4 g/kg in minimally-pressed rapeseed oil was observed by Siger et al. [34]. Cai and Arntfield [35] recorded different concentration of sinapic acid in canola flour, industrial meal, and a protein isolate as 0.39, 0.24, and 0.20 mg/g, respectively. Szydłowska-Czerniak et al. [36] identified sinapic acid as the main phenolic acid in rapeseed varieties with concentrations in the 17.4−36.4 mg/100 g range.

The concentration of canolol in the defatted meal increased progressively with an increase in seed air frying temperature-time regimens and ranged from 84.92 ± 3.41 to 151.92 ± 2.73 µg/g DW (Table 1). The maximum canolol contents (15.35 ± 7.7, 149.59 ± 5.70 and 151.62 ± 2.80 µg/g DW) were obtained at air frying temperatures of 170, 180 and 190 °C for 20 min, respectively, with no significant (*p* > 0.05) differences between these temperatures (Table 1). These values are significantly (*p* < 0.05) lower than those reported by Nandasiri et al. [7]). Their maximum concentration obtained for canolol was 427.11 ± 7.12 μg/g DW at 180 °C for 5 min and 453.4 ± 17.66 μg/g DW at 160 °C for 10 min in canola meal treated in an inert pressurized heating chamber of RapidOxy 100. Zago et al. [16] reported the highest canolol content (0.12–0.18 mg/g DW) when canola was subjected to a simple roasting treatment at 160 °C, although the highest canolol content (0.40 mg/g/DW) was recorded following microwave treatment (2.45 GHz and 2 kW) for 5 min with no interval, which is similar to the findings of Nandasiri et al. [7], but different from our current findings with the air-frying treatment. Khattab et al. [25] also exposed rapeseed to microwave treatment (power 3.7 kW) for 13 min and obtained a 58% conversion rate in canolol (4.16 mg/g DW). The amount of canolol recovered from the canola meal was significantly lower than the corresponding levels of sinapine and sinapic acid (Table 1). This may be attributed to little or no breakdown of sinapine to sinapic acid and/or the lack of breakdown and decarboxylation of bound sinapic acid to canolol under the elevated temperature of air frying. Moreover, the low concentration of canolol observed in our current study may also be attributed to the lipophilic nature of canolol and its high solubility in oil-enriched products. The alteration in the chemical structures of phenolics through the breakdown of various bonds while impacting the cellular matrix is influenced by processing at elevated temperatures [37]. Sorensen et al. [38] showed that canolol and other phenolics could be extracted from defatted meal at higher temperatures (>185 °C). Wang et al. [27] obtained canolol yield (40.74 mg/kg and 9.66 mg/kg), respectively, in the oil fractions of *B. napus* and *Brassica rapa* rapeseed varieties pre-treated using steam explosion. The authors recorded improved canolol content. However, there was a reduction in the quantity of other phenolic derivatives. This observation is consistent with the recovery of canolol and other SADs, as previously reported [7,28]. Yang et al. [28] further reported a maximum yield of canolol in rapeseed after 7 min (89.89 mg/100 g) of microwave pre-treatment, which was 6 times greater than the level in unroasted rapeseed (14.07 mg/100 g). Our current findings are consistent with previous studies showing that the recovery of individual phenolic compounds was dependent on various temperature-time combinations.

### 3.2. Impact of Air Frying of Seeds on the TPC and Antioxidative Properties of Defatted Canola Meal Methanolic Extracts

#### 3.2.1. Total Phenolic Content (TPC)

Changes in the TPC of canola meal as affected by air frying are shown in Figure 2.

A few studies documenting the impact of seed roasting by different pre-treatment methods on TPC have been reported [10,16,18,28,39]. The highest TPC values (3.15 ± 0.14 and 3.05 ± 0.02 mg GAE/g DW) were observed in canola meal extracts after air frying at 190 °C for 15 and 20 min, respectively, with no significant difference (*p* > 0.05) between these two times (Figure 2). In contrast, the lowest TPC value in defatted meal (0.88 ± 0.04 mg GAE/g DW) was obtained when the seeds were air fried at 160 °C for 5 min. In general, the TPC increased with increases in air frying temperature and time combinations (Figure 2). There was a significant and positive correlation between the TPC and canolol content (r = 0.75, *p* < 0.05). However, a significant (*p* < 0.05) and negative association was observed between TPC and sinapine (r = −0.56) as well as TPC and sinapic acid (r = −0.91) (Table 2). Zago et al. [16] subjected rapeseed meal to simple roasting and superheated steam treatments at 105 °C and 160 °C and obtained the highest TPC value of 17.40 mg/g DW. They also observed 11% and 27% reductions in the TPC contents after roasting and superheated steam treatment, respectively. The decreases in TPC values were attributed to the thermally-induced degradation of phenolic compounds. However, our current findings are in contrast as increases in roasting temperature and time led to higher TPC up to the maximum air frying temperature and time regimens. Cai and Arntfield [35], in their study, obtained a TPC value of 19.90 mg/g DW in rapeseed. Nandasiri et al. [10] and Li and Guo [9] reported the highest TPC values of 20.71 mg SAE/g DM and 92.53 mg SAE/g meal in canola/rapeseed meal using accelerated solvent and pressurized solvent extraction techniques, respectively. The highest TPC values were obtained by both authors at 180 °C and 200 °C process temperatures, respectively. Wang et al. [27] reported that TPCs in *B. rapa* and *B. juncea* rapeseed increased significantly by 31% and 67%, respectively, while there was a decline in the TPC in *B. napus* rapeseed exposed to steam explosion treatment. In the same study, the authors concluded that there was a significant increase in TPCs in the rapeseed product, cakes, and defatted cakes of *B. rapa* and *B. juncea* following steam explosion treatment. In a study by Szydłowska-Czerniak et al. [36], the estimated total phenolic acids in rapeseed flours from HPLC data ranged from 20.3 mg to 40.7 mg/100 g, while sinapic acid, the major phenolic acid, had values, which varied from 17.4 to 36.4 mg/100 g.

#### 3.2.2. 2,2-Diphenyl-1-picrylhydrazyl (DPPH) Radical Scavenging Activity

The impact of seed roasting by air frying on the DPPH free radical scavenging capacity of defatted canola meal methanolic extracts is shown in Figure 3. The canola meal extracts were capable of effectively scavenging the DPPH radical. The canola meal extract obtained at 180 °C for 5 min seed roasting showed the highest DPPH scavenging activity (80%), while the least activity (73%) was observed in extracts from seeds that had been air fried at 190 °C for 20 min. Generally, all the methanolic extracts showed high radical scavenging activity with significant (*p* < 0.05) differences between the values. Moreover, a strong and significant (*p* < 0.05) correlation was found between the DPPH radical scavenging activity of the extracts and sinapine (r = 0.63), sinapic acid (r = 0.67), RT = 26.6 min compound (r = 0.62), and canolol (r = −0.63) (Table 2). There was a strong positive and significant correlation between DPPH radical scavenging activity and ferric reducing antioxidant power (r = 0.70) and metal ion chelation activity (r = 0.70). Conversely, a significant (*p* < 0.05) and moderate negative correlation (r = −0.54) was found between DPPH and TPC of the extracts obtained at various air frying treatments (Table 2)**.** The ability of rapeseed crude oil phenolics to effectively scavenge DPPH free radicals through synergistic action was reported by [40]. The authors observed a stronger antioxidant activity for a mixture of sinapic acid, canolol, and sinapine compared to individual phenolic compounds. Teh et al. [41] reported DPPH scavenging values ranging between 33.2–38.8% in extracts of canola seed cake obtained via microwave and PEF-assisted extractions. Wang et al. [27] studied the effects of steam explosion on *B. napus* rapeseed and its products, and the highest antioxidant activity using the DPPH method was obtained for defatted rapeseed cake (5540 μmol Trolox equivalent (TE)/100 g) and rapeseed cake (4948 μmol TE/100 g).

#### 3.2.3. Ferric Reducing Antioxidant Power (FRAP)

The ability of phenolic compounds to reduce ferric ions to form stable ferrous ions isused as an indicator of the ability to mitigate oxidative stress. The FRAP values of defatted canola meal methanolic extracts obtained from seeds roasted at various roasting conditions are shown in Figure 4.

The highest FRAP value (0.53 ± 0.04 mM) was observed in the defatted canola meal extract from seeds air fried at 160 °C for 5 min, while the lowest values (0.31 ± 0.01 and 0.32 ± 0.01) were observed in meal extracts from seeds subjected to air frying at 170 and 190 °C for 20 min, respectively (Figure 4). Generally, a downward trend was apparent for the FRAP values with an increase in air frying temperature and time combinations. The study by Wang et al. [27] showed that rapeseed products treated with steam explosion had FRAP values of 6944 μmol TE/100 g and 7432 μmol TE/100 g in *B. rapa* defatted rapeseed cake and *B. juncea,* respectively. In the same study, they also reported a FRAP value (1366.5 μmol TE/100 g) in oil produced from pretreated *B. napus* rapeseed, which was greater than that of untreated *B. napus* rapeseed (0.51 μmol TE/100 g). It was concluded that steam explosion treatment significantly (*p* < 0.05) improved the antioxidant activity of *B. napus* rapeseed oil. Teh et al. [40] observed FRAP values between 22.04–23.45 μmol Fe (II)/g in canola seed cake extracts obtained from microwave and pulsed electric field-assisted extractions. The FRAP value obtained in our current study was significantly (*p* < 0.05) negatively correlated with canolol (r = −0.83) (Table 2). The observed negative correlation between canolol and FRAP suggests that canolol may be effective in reducing ferric ions to more stable ferrous ions. However, there was a strong positive and significant (*p* < 0.05) correlation between FRAP and sinapine (r = 0.91), sinapic acid (r = 0.81), and the RT-26.6 min compound (r = 0.81) (Table 2). It is suggested that the improved antioxidant activity of the extracts in terms of FRAP could be largely due to the contributory effects of the predominant phenolic compounds- sinapine, sinapic acid and the RT-26.6 min compound present in the air-fried canola meal extracts. The improved antioxidant activities of the canola meal extracts evaluated by the FRAP method may also be attributed to some Maillard reaction products, which are formed at elevated temperatures [42,43]. Moreover, a strong positive correlation and significant (*p* < 0.05) relationship also existed between FRAP and DPPH (r = 0.70) as well as FRAP and metal ion chelation (r = 0.81) activities of the extracts. However, FRAP was significantly (*p* < 0.05) negatively correlated with TPC (Table 2), as can be discerned from a comparison of Figure 2 and Figure 4.

#### 3.2.4. Metal Ion Chelating (MIC) Activity

The metal-chelating ability of a compound is important in the food industry as it helps to reduce the concentration of free metal ions, which are promoters of lipid peroxidation. Moreover, metal chelating agents are regarded as secondary antioxidants as they help to reduce redox potential, thereby stabilizing the oxidized metal ion [44]. The chelating ability of the various canola meal methanolic extracts produced from seeds air fried at different temperature-time combinations are shown in Figure 5.

The highest activity (75%) was observed in meal extracts obtained from seeds that were air fried at 160 °C for 20 min, which is a strong indicator of iron-binding potential. Conversely, the least metal ion chelating activity (74%) was obtained in meal extracts from seeds pre-treated at 190 °C and 20 min (Figure 5). The effectiveness of chelating agents as potent antioxidants has been attributed to their capacity to stabilize oxidized metal ions through reduction of the redox potential [45]. A significant and positive correlation was evident between the metal ion chelation property and sinapine (r = 0.80), sinapic acid (r = 0.70), and the RT-26.6 min compound (r = 0.80). These results suggest that the improved antioxidant activity exhibited by the extracts in terms of chelation of metal ions was due to the synergistic interactions between the major sinapic acid derivatives recovered in the canola meal extracts. In addition, a significant (*p* < 0.05) and positive correlation existed between metal ion chelation properties and DPPH (r = 0.70) as well as FRAP (r = 0.85) for all the extracts (Table 2). The strong correlations observed between these antioxidant assays (FRAP, MIC and DPPH) reflect the potent antioxidant efficacy of the extracts obtained under the different air frying conditions.

## 4. Conclusions

The current study investigated the influence of air frying pre-treatment of canola seeds on the formation of canolol and other sinapates, which can be extracted from defatted canola meal. The findings demonstrated the potential valorization of residual canola meal via seed roasting by air frying as a green pre-treatment approach to generate the major sinapic acid derivatives and improve the antioxidative potential of the extracts. The highest contents of the major sinapates, i.e., sinapine, sinapic acid and the RT-26.6 min compound, were obtained by air-frying the seeds at 160 °C for 5 min, while the highest canolol concentration was from the 170 °C for 20 min air frying condition. The canola meal extracts obtained at various air frying conditions showed improved TPC and exhibited enhanced in vitro antioxidant potential with respect to free radical scavenging, ferric ion reduction to ferrous ion and chelation of metal ions, all of which are implicated in lipid peroxidation. For the optimum recovery of canolol, an air frying condition of 170 °C for 20 min is recommended, while for other sinapates viz- sinapine, and sinapic acid, an air frying condition of 160 °C for 5 min is recommended. The outcome of this study will further contribute to the canola industry’s quest for value-added processing of canola seeds to generate natural antioxidants with potential applications in the feed, cosmetics, and pharmaceutical industries.

## Figures and Tables

**Figure 1 antioxidants-11-02411-f001:**
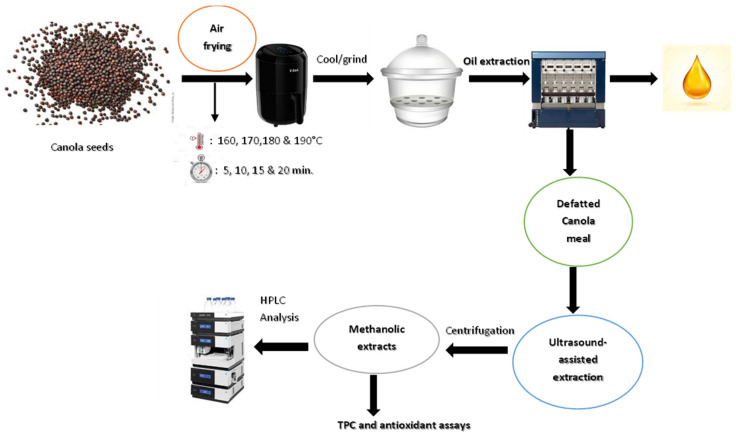
Extraction of phenolic compounds from de-oiled canola meal through air frying pre-treatment and ultrasound-assisted extraction.

**Figure 2 antioxidants-11-02411-f002:**
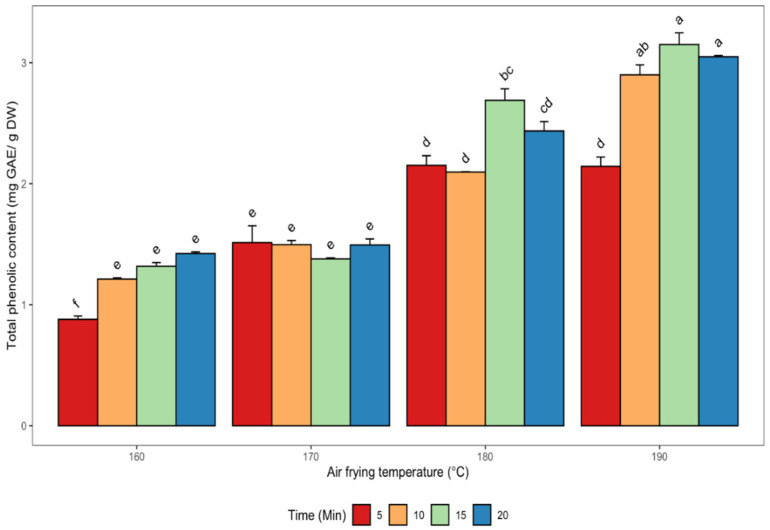
Changes in the total phenolic content of de-oiled canola meal extracts as affected by different air frying conditions. Error bars indicate the SD of the measurements for each sample (*n*  =  3). Bars with different letters (a–f) have significantly (*p* < 0.05) different mean values.

**Figure 3 antioxidants-11-02411-f003:**
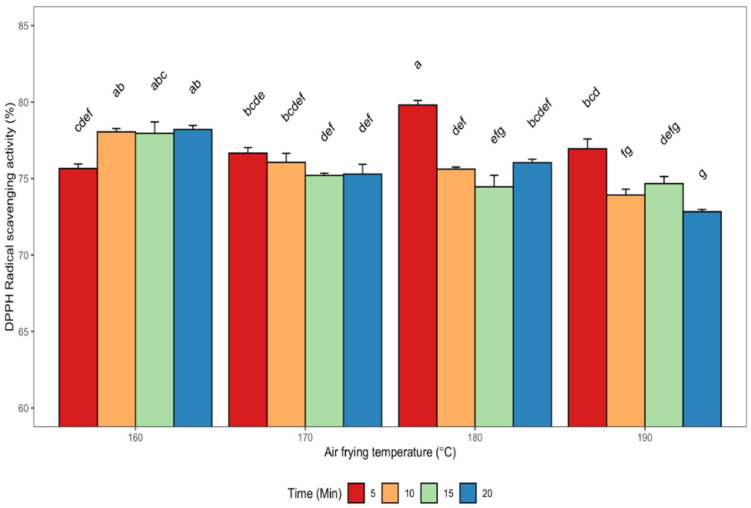
Changes in the DPPH free radical scavenging ability of de-oiled canola meal extracts as affected by different air frying conditions. Error bars indicate the SD of the measurements for each sample (*n * =  3). Bars with different letters (a–g) have significantly (*p* < 0.05) different mean values.

**Figure 4 antioxidants-11-02411-f004:**
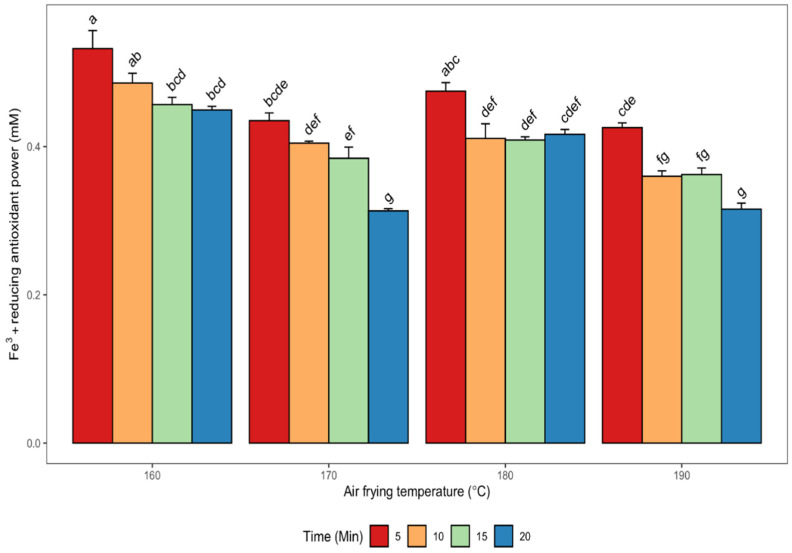
Changes in ferric reducing antioxidant power (FRAP) of de-oiled canola meal extracts as affected by different air frying conditions. Error bars indicate the SD of the measurements for each sample (*n*  =  3). Bars with different letters (a–g) have significantly (*p* < 0.05) different mean values.

**Figure 5 antioxidants-11-02411-f005:**
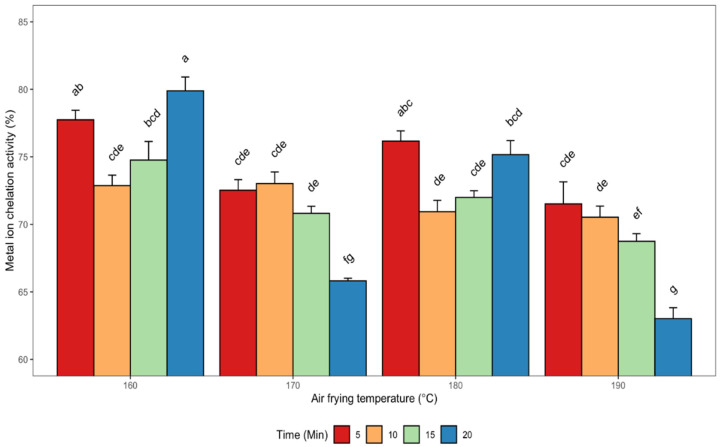
Changes in metal ion chelation activity (MIC) of de-oiled canola meal extracts as affected by different air frying conditions. Error bars indicate the SD of the measurements for each sample (*n*  =  3). Bars with different letters (a–g) have significantly (*p* < 0.05) different mean values.

**Table 1 antioxidants-11-02411-t001:** Effect of air frying pre-treatment on the recovery of major sinapic acid derivates (SADs) in de-oiled canola meal. Canola meal-derived sinapic acid derivatives (SADs).

Air Frying Condition	Sinapine (µg SPE/g DW)	Sinapic Acid (µg SAE/g DW)	Canolol (µg CE/g DW)	RT-26.6 min (µgSPE/g DW)	RT-32.4 min (µg SPE/g DW)	RT-27.28 min (µg SPE/g DW)	RT-23.81 min (µg SPE/g DW)
160/5 min	7572 ± 479 ^a^	727 ± 44 ^a^	84.92 ± 3.41 ^i^	1763 ± 74 ^a^	903 ± 26 ^efg^	375 ± 19 ^de^	458 ± 19 ^ab^
160/10 min	7471 ± 198 ^a^	646 ± 22 ^b^	92.77 ± 0.88 ^hi^	1681 ± 50 ^abc^	909 ± 44 ^ef^	381 ± 22 ^cde^	430 ± 22 ^bcde^
160/15 min	7187 ± 119 ^ab^	632 ± 46 ^bc^	96.81 ± 3.96 ^gh^	1678 ± 81 ^abc^	1103 ± 56 ^cd^	381 ± 23 ^cde^	406 ± 7 ^def^
160/20 min	7193 ± 86 ^ab^	603 ± 18 ^bcd^	109.17 ± 1.61 ^def^	1753 ± 22 ^ab^	1077 ± 17 ^cd^	406 ± 4 ^abcd^	448 ± 6 ^abc^
170/5 min	6772 ± 209 ^bc^	644 ± 72 ^b^	97.56 ± 0.08 ^fgh^	1668 ± 75 ^abcd^	1111 ± 33 ^bcd^	394 ± 17 ^abcd^	427 ± 16 ^bcde^
170/10 min	6649 ± 110 ^cd^	598 ± 29 ^bcd^	103.03 ± 1.53 ^efg^	1650 ± 53 ^abcd^	1233 ± 73 ^ab^	401 ± 27 ^abcd^	440 ± 18 ^abcd^
170/15 min	6231 ± 105 ^def^	563 ± 28 ^cde^	121.38 ± 6.25 ^c^	1576 ± 23 ^cde^	1305 ± 32 ^a^	393 ± 12 ^bcd^	410 ± 14 ^def^
170/20 min	5417 ± 114 ^g^	422 ± 21 ^gh^	151.35 ± 7.65 ^a^	1359 ± 27 ^f^	1161 ± 64 ^bc^	348 ± 19 ^e^	376 ± 18 ^f^
180/5 min	6791 ± 124 ^bc^	543 ± 3 ^def^	113.60 ± 1.26 ^cde^	1672 ± 30 ^abcd^	801 ± 55 ^fg^	438 ± 9 ^a^	472 ± 10 ^a^
180/10 min	6066 ± 19 ^ef^	477 ± 27 ^fg^	115.55 ± 2.12 ^cd^	1458 ± 8 ^ef^	824 ± 39 ^fg^	380 ± 6 ^cde^	404 ± 12 ^ef^
180/15 min	6208 ± 94 ^def^	403 ± 15 ^h^	134.76 ± 2.32 ^b^	1485 ± 38 ^ef^	796 ± 17 ^g^	387 ± 8 ^cd^	411 ± 11 ^cdef^
180/20 min	6559 ± 42 ^cd^	374 ± 4 ^h^	149.59 ± 5.70 ^a^	1584 ± 9 ^cde^	806 ± 15 ^fg^	417 ± 5 ^abc^	449 ± 8 ^abc^
190/5 min	6526 ± 33 ^cde^	523 ± 11 ^ef^	100.05 ± 0.72 ^fgh^	1636 ± 16 ^bcd^	994 ± 17 ^de^	428 ± 4 ^ab^	472 ± 4 ^a^
190/10 min	6309 ± 56 ^cdef^	360 ± 11 ^hi^	139.75 ± 3.04 ^b^	1578 ± 423 ^cde^	898 ± 44 ^efg^	414 ± 8 ^abcd^	465 ± 14 ^ab^
190/15 min	6241 ± 130 ^def^	295 ± 19 ^i^	151.92 ± 2.73 ^a^	1552 ± 44 ^de^	876 ± 41 ^efg^	439 ± 4 ^a^	470 ± 20 ^a^
190/20 min	5940 ± 115 ^f^	214 ± 17 ^j^	151.62 ± 2.8 ^a^	1489 ± 31 ^e^	895 ± 38 ^efg^	414 ± 12 ^abcd^	466 ± 10 ^a^

Means with different letter superscripts in each column are significantly different (*p* < 0.05) from each. Values are mean ± SD of three replicates (*n* = 3). RT- retention time, SPE- sinapine equivalent, SAE- sinapic acid equivalent, CE- canolol equivalent, DW- dry weight.

**Table 2 antioxidants-11-02411-t002:** Pearson correlation matrix between canola meal-derived sinapates, total phenolic content and antioxidant activity.

	Canolol	Sinapine	Sinapic Acid	RT-26.6 min	RT-32.4 min	RT-27.38 min	RT-23.81 min	DPPH	FRAP	MIC	TPC
Canolol	1										
Sinapine	−0.79 ***	1									
Sinapic Acid	−0.93 ***	0.75 ***	1								
26.6 RT	−0.75 ***	0.93 ***	0.72 **	1							
32.4 RT	−0.25	−0.04	0.40	0.10	1						
27.38 RT	0.21	0.03	−0.37	0.25	−0.35	1					
23.81 RT	0.00	0.29	−0.19	0.46	−0.42	0.85 ***	1				
DPPH	−0.63 **	0.63 **	0.67 **	0.62 *	0.07	0.08	0.03	1			
FRAP	−0.83 ***	0.91 ***	0.81 ***	0.81 ***	−0.17	−0.04	0.17	0.70 **	1		
MIC	−0.63 **	0.80 ***	0.70 **	0.80 **	−0.05	0.07	0.17	0.70 **	0.85 ***	1	
TPC	0.75 ***	−0.56 *	−0.91 ***	−0.49	−0.57 *	0.62 *	0.43	−0.54 *	−0.59 *	−0.51 *	1

DPPH 2,2-Diphenyl-1-picrylhydrazyl, FRAP Ferric reducing antioxidant power, MIC Metal ion chelation, TPC Total phenolic content. RT Retention time, RT-26.6 min, RT-32.4 min, RT-27.38 min, RT-23.81 min refers to unidentified compounds at different RT. *** Very highly significant at *p* < 0.001, ** Highly significant at *p* < 0.01, * Significant at *p* < 0.05.

## Data Availability

Data supporting the findings of this study are available from the corresponding authors upon reasonable request.

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
