# Peer review of "Improved Extraction Efficiency and Antioxidant Activity of Defatted Canola Meal Extract Phenolic Compounds Obtained from Air-Fried Seeds"

_antioxidants, 2022, doi:10.3390/antiox11122411_

Round 1

Reviewer 1 Report

This paper describes a improved extraction efficiency and antioxidant activity of defatted canola meal extracts obtained from air-fried seeds. The article is quite complete, it is of interest to the scientific community, the methods and statistics used are appropriate and the results and discussion are conveniently described. The work is well discussed and is supported by the references provided by the authors. The English language is correct. The work is interesting and delves in the extraction efficiency of defatted canola meal.

I consider that the article is appropriate to be published in Antioxidants journal once the authors have made some modifications to it.

Major aspects:

Materials and methods: Include a "Reagent Section". In this section, authors should describe all reagents and solvents used.

Materials and methods: In the different sections, although the authors refer to previously established methods, they must indicate at least briefly the equipment used, calibration curves, r2, range of linearity of each of the methods employed, standards employed.

Minos aspects:

Title: Capitalize each word according the format of the journal.

Lines 30, 103, 152, 314 ….: Put a separation after and before “=”,  “±”. Unify and apply to the entire document.

Line 38: Put “Brassica napus” in italics.

Lines 52, 168, 171,…..: Check the format of  “ºC”. Unify and apply to the entire document.

Lines 70, 73, 74, 195,……: Indicate name et al [x]. Unify and apply to the entire document.

Lines 89, 91, 94,…..: Capitalize each word according the format of the journal. Unify and apply to the entire document.

Lines 103, 124, 141, …: Put “n” in italics. Unify and apply to the entire document.

Line 109: “assisted”?.

Line 119: “96-well”.

Lines 128, 140,…: Put “p” in italics. Unify and apply to the entire document.

Lines 177, ….: Do not put a separation between a number and “%”. Unify and apply to the entire document.

References: Check the correct format of the names of the journals. All abbreviated, and use a “.” After each abbreviation.

Reviewer 2 Report

The manuscript reports very interesting results regarding the influence of air frying pre-treatment of canola seeds on the recovery of important bioactives (sinapates) by ultrasound-assisted extraction with 70% (v/v) methanol from defatted canola meal, and on the formation of canolol.

The manuscript is well structured, the text is clearly presented and will be of interest to the Antioxidants readers.

Hence, it merits publication provided minor revisions and editing are introduced, namely:

Line 6:  The affiliation of the last author - should be Scalnlon1,2

Lines: 70, 171, 174, 178, 195, etc. A sentence cannot begin by a Reference, should be revised accordingly in all places through the manuscript.

Lines 55-56 –The sentence should be either revised or parts of it combined with the sentence on lines 53-54 to avoid repetitions.

Table 1, line 140: Means with different letter subscript in each column… There are no letter subscripts?

A comment and a subsequent question:

It is well known that many phenolic compounds are oxidized at higher temperatures. It was even argued that temperatures above 343.15 K are particularly harmful since fast polyphenol degradation could be triggered.

Consequently, the observations and comments on Lines 136, 137; 146-147; 159-160, etc. come as no surprise. On the other hand, higher temperatures have generally a positive effect on the TPC (confirmed in the present study, see lines 24-25).

In view of the above,

Have the authors considered lowering the temperature of the seeds pretreatment for example to 150 C?

As the influence of temperature on a phenolic compound sensitivity is not straightforward and depends on many factors, among those its type, physicochemical and biochemical characteristics, and lastly, but very importantly, on the nature of the plant matrix, it would have been very interesting to have information on which is the minimum pretreatment temperature that will guarantee a high content of sinapic acid derivatives and a reasonable amount of canolol? 

Reviewer 3 Report

The manuscript entitled “Improved extraction efficiency and antioxidant activity of defatted canola meal extract phenolic compounds obtained from air-fried seeds” is interesting and suitable for publication in Antioxidants after minor revision.

In the reviewer’s opinion, there are a few things to improve before acceptation

-        it is absolutely necessary to complete the material and methods chapter. Extraction conditions, and type of equipment used, especially since in many cases the authors state that the method has been modified. This should be described in detail.

-        the tables should be formatted correctly, the lack of order makes it very difficult to analyze the results. The result should fit in one line, and the statistical analysis can be (letters, asterisks) in the next. If a unit refers to several compounds, it can be stated collectively

-        in chapter 2.1 you need to add a reference (number).

-        Citations in the text should not start with the item number in the list, but rather with the author's surname and then the item number.

Round 2

Reviewer 1 Report

The authors have made the indicated modifications and the article has improved substantially. For this reason, I consider that the article can be considered for publication in Foods journal in its current form.

Anyway I suggest small additional modifications prior to publication, that can be done during the proofreading of the manuscript.

Line 6: I think an author is missing.

Title, lines 88, 89, etc..: Articles and prepositions do not need to be capitalized according the format of the journal. Unify and apply to the entire document.

Lines 92, 106, …..: Include the city of all the companies cited. In case of USA companies, include the city and the state abbreviation. Unify and apply to the entire document. Include the company of the canolol, sinapic acid and sinapine.